# Human Capital, Market Environment, and Firm Innovation in Chinese Manufacturing Firms

**Xiuli Sun**

School of Statistics, Southwestern University of Finance and Economics, Chengdu 611130, China;
sunxl@swufe.edu.cn

**Abstract:** This paper explored firm-level innovation in different market environments from a human capital point of view using both theoretical and empirical approaches. In the theoretical model, two firms compete with each other in a two-stage Cournot competition game, the innovation stage and production stage. Theoretical results indicated that a firm's innovation is not only determined by its human capital level, firm characteristics, and its market share, but also might be affected by market environment. In the empirical study, we used two firm-level datasets from China, one from metropolitan cities and one from mid-sized cities. Results show that skilled human capital is vital for firm innovation in metropolitan cities, while R&D plays an important role in firm innovation in mid-sized cities. The GM's experience is more important for firms in metropolitan cities, while the GM's education is more critical for firms in mid-sized cities. Moreover, GDP per capita has a positive effect on firm innovation in metropolitan cities, while it harms firm innovation in mid-sized cities. We further showed that the industry composition explanation could account for our results. Finally, we also tried IV estimation, and the related results are discussed.

**Keywords:** human capital; market environment; education; R&D; innovation

## 1. Introduction

Why do firms differ in innovation? Economists have long sought answers to this question because the characteristics of innovative firms can have significant implications, not only for firm success but also for the economic growth of a country. Along this avenue, the literature on testing Schumpeterian hypotheses, which argue that large firms operating in a concentrated market are the main engine of technological progress, has offered numerous insights into how firm size and market structure affect firm innovation. Although it provides a practical framework for exploring the issue, the large body of work also leaves many questions unanswered. Most notably, the literature, which almost exclusively focuses on firm size, industry characteristics, market structure, and/or human capital information, seldom touches on the characteristics of firms beyond size and strategic choice. Moreover, leaving strategic choices outside the framework results in ignoring the interaction between a firm and its market environment.

The spatial differences in the market environment in China are huge. In 2021, Beijing achieved the highest GDP per capita, and it was 184 thousand Yuan (around 28.9 thousand dollars), and the lowest province, Gansu, only attained 41 thousand Yuan. Traditionally, the gap can be explained by natural advantages, labor sorting, and agglomeration economies, among others. Meanwhile, recent firm selection theory argues that larger markets feature tougher competition, leading less productive firms to exit. To survive, firms have to make their best innovation strategy in a specific market environment. Thus, firms with similar internal resources in different market environments might have different innovation strategies [1–3].

Traditionally, the most important explanatory factor of innovation is R&D spending because it is believed that R&D is the input in producing innovation. However, in essence,

those studies are deficient. First, they simply regard R&D as the most crucial input of innovation without going deeper into the details of R&D and the mechanisms of R&D in affecting innovation. Thus, they fail to take other firm resources, mainly a firm's skilled human capital, into consideration. Second, they ignore non-R&D innovation, which is usually vital to firm innovation.

In addition, innovation includes not only R&D innovation but also non-R&D innovation. Generally, there are three types of creative activities that do not require R&D. First, many imitative activities, including reverse engineering, do not require R&D investment, and imitation mainly depends on the firm's technical personnel and engineers. Second, firms can make minor modifications or incremental changes to products and processes, relying on engineering human capital. Moreover, researchers noted that the innovation process in low-and medium-technology sectors is more related to adaptation and learning by doing, based on design and process optimization, rather than on R&D [4]. Third, firms can combine existing knowledge in new ways, for example, in industrial design and engineering projects [5]. Due to the large share of firms that innovate without performing R&D, we argue that studies that only focus on R&D should not be enough to explain innovation differences across firms fully.

Recently, scholars have noticed that the institutional and competitive contexts in which a firm operates will affect the relationship between human capital and innovation [1]. Different market environments exhibit significant heterogeneity in terms of labor regulations [6], innovative infrastructure [7], and institutional quality [8]. The efficiency of the internal factors, such as human capital and R&D, to promote firm innovation will be conditioned by the market environment, i.e., the economic and institutional characteristics of the environment in which a firm operates [9].

In sum, the paper falls into three different strands of the literature: (1) the literature on testing Schumpeterian hypotheses, such as Hong et al. and Raymond et al. [10,11]; (2) the relationship between market environment and firm innovation, such as Krammer, Özen and Baycan, and Qiu and Wang [1–3]; and (3) the relationship between human capital and firm innovation based on resource-based theory, endogenous growth theory, and upper echelon theory, such as Custódio et al., Romer, Gennaioli et al., Squicciarini and Voigtländer, Lin et al., McGuirk et al., and Zhang et al. [12–18].

This paper explores the relationship between human capital and firm innovation as well as the market environment that can affect this relationship. In the theoretical model, two firms compete with each other in a two-stage Cournot competition game, the innovation stage and production stage. Skilled human capital can affect innovation success probability directly and, via R&D level, indirectly. Managerial human capital can affect firm innovation through their choice of projects and R&D level. We find that firm innovation is not only determined by its human capital level, firm characteristics, and its market share, but also might be affected by the market environment. In the empirical study, we use two firm-level datasets from China, one from metropolitan cities and one from mid-sized cities. Human capital indicators are skilled human capital (number of highly educated workers), the general manager (GM)'s education and experience, and the management team's education and age. We find that skilled human capital is vital for firm innovation in metropolitan cities, while R&D plays an important role in firm innovation in mid-sized cities. The GM's experience is more critical for firms in metropolitan cities, while the GM's education is more vital for firms in mid-sized cities. The management team's education tends to have a positive effect on firm innovation, while the team's average age has a negative and significant effect on firm innovation. Moreover, GDP per capita has a positive effect on firm innovation in metropolitan cities, while it hurts firm innovation in mid-sized cities. The reason behind this is the industry composition of a city.

The endogeneity of skilled human capital may bias our estimates. We use the instrumental variable method to solve this problem. The instruments we used are the number of job applicants for skilled worker positions, the number of weeks to fill the last job, the percentage of redundancy in unskilled workers, and the city-level average wage.

This paper makes three main contributions. (1) This paper explores firm-level innovation in different market environments from a human capital point of view using both theoretical and empirical approaches. (2) Using detailed firm-level data, we could study the effects of skilled human capital, the general manager's education and experience, and the management team's education and age. (3) Two datasets from two different levels of cities, metropolitan cities and provincial middle cities, enable us to examine the effect of the market environment on firm innovation. We also include GDP per capita to examine the effects of different economic development on firm innovation.

The paper is organized as follows. Section 2 introduces firm-level human capital into innovation. Section 3 presents a theoretical framework where two firms Cournot compete with each other in a two-stage game. In Section 4 we introduce the data. Section 5 introduces our methodology strategy. In Section 6, we present our main results and discuss the findings. Section 7 presents further investigation. Section 8 concludes.

## 2. Human Capital and Innovation

Why is human capital essential in the study of firm innovation? According to the resource-based view of the firm, performance differences across firms can be attributed to variance in the firms' resources and capacities. Resources that are valuable, unique, and difficult to imitate can provide the basis for firms' competitive advantages. Among all the resources in a firm, human capital has long been argued as a critical resource [12,19]. Although human resources may be mobile to some degree, because some capabilities are based on firm-specific knowledge (for example, some capabilities about how to collaborate efficiently with a certain colleague), and others may only be valuable when integrated with additional individual capacities and specific firm resources (for example, complementary assets) that may not be mobile (for example, scientists need some certain experimental equipment to be productive) [19], the idea that a firm's human capital is critical still holds. Moreover, the upper echelon theory argues that organizations are just reflections of their top managers [20]. Thus, given the importance of firm human capital, studying firm innovation from a human capital view becomes natural.

On a macro scale, human capital has long been introduced into innovation in endogenous growth theory [12,13]. Other scholars further proposed that higher human capital stock tends to generate higher growth through at least two channels [21]: on the one hand, more human capital facilitates the absorption of superior technologies from leading countries, and for this channel, schooling at secondary and higher levels should be vital; on the other hand, human capital tends to be more difficult to adjust than physical capital. The endogenous growth theory takes human capital as one of the most critical inputs in innovation at the macro level [13,22], and this inspires us to notice the importance of human capital in firm innovation. However, we still know relatively little about firm-level human capital and innovation, given the difference between micro and macro studies.

Firm-level human capital can be divided into two types: managerial human capital and skilled human capital. Managerial human capital is embodied in CEOs and top management teams. Top executives have the discretion to control R&D expenditure in firms. Furthermore, because R&D expenditure is a long-term investment that is considerably risky with high failure rates, top managers monitor R&D expenditure closely and adjust its level based on their preferences. Moreover, top management teams have the task of formulating and implementing the firm's strategy [20], and as part of their leadership function, CEOs must coordinate and control team behaviors.

On the other hand, skilled human capital is related to all skilled workers in a firm, and can be seen as a general measure of human capital in a firm, and thus it is fundamental to a firm's behavior and performance. The mechanisms between a firm's skilled human capital and innovation can function in two channels. First, higher-skilled human capital means a higher ability of learning by doing and thus can improve a firm's innovation ability. Scholars have studied the relationship between learning by doing and patents, and they found that patenting in process innovation in the chemical industry was largely an

outgrowth of "learning by doing." Second, skilled human capital and a firm's R&D together affect the firm's innovation through R&D innovation. The complementary relationship is modeled by Romer [13], where innovation is produced by combining R&D and human capital together.

### 3. Theoretical Framework

This section presents our theoretical framework, and allows us to see how firms choose their optimal innovation project, optimal human capital, and R&D investment. Thus, we can derive expressions for firm innovation, which we estimate in the empirical part.

Our framework is based on previous studies [23,24]. We examined firm innovation using a duopoly model with risk-neutral firms in a three-stage game. Firms engage in innovation because a successful project lowers their production cost in the sequent output market competition. To simplify our analysis, we used cost-reducing technology to represent innovation because we can always break a product into a Lancasterian bundle of services and model product improvement as a reduction in the cost of producing services. In the first stage, the firms invest in a risky R&D project. In the second stage, they will choose their own human capital level. In the third stage, the firms choose their R&D level.

Since backward induction can give us subgame-perfect equilibrium, we consider first the output market decision in the production stage. We considered an industry consisting of two firms with Cournot competition. The firms produce a single homogenous good, and each maximizes its single-period profit. Assume the expected output market profit be a function of the firms' constant marginal costs of production $c_i$, $i = 1, 2$. The inverse demand curve the firms face is linear. The single period profits of the $i$th firm are given by

$$\Pi_i = \left[A - 2c_i + c_j\right]^2 \tag{1}$$

where $A$ is subject to $A - 2c_i + c_j > 0$ for $i, j \in \{1, 2\}$. It can be seen as the total demand of a certain market and other market-specific factors. Equation (1) implies that firm $i$'s profit is decreasing with its own cost but increasing with its rival's cost, and thus the firm's profit will increase with its own innovation success but decrease with its rival's innovation success. Therefore, both $A$ and $c_j$ will capture the effects of the market environment.

In the innovation stage, the two firms choose their own R&D project, human capital level, and the level of R&D investment sequentially. Moreover, we assume that they conduct their own projects simultaneously and are completed before the start of the output game. Two outcomes may arise for each project: either it succeeds or fails. The set of R&D strategies from which the firms choose and the innovation outcome is common knowledge.

There are three substages in the innovation stage. First, both firms choose projects $p^i$ ($i = 1$, 2) from the continuum of projects, $\alpha$, in the set $(0, 1)$. Higher values of $\alpha$ represent projects with a greater chance of success at any fixed level of investment. If a project $\alpha$ yields a successful innovation for a firm, then the firm's cost is reduced by $\gamma(\alpha)$, where $\gamma$ is differentiable in $\alpha$ and $\gamma'(\alpha) < 0$, which means that as $\alpha$ increases from 0 to 1, the cost reduction will decrease. Therefore, if firm $i$ succeeds in innovation, its marginal cost will become $c_i - \gamma(\alpha_i)$, and if it fails, its marginal cost will still be $C_i$. Projects should be done based on the existing technology base, the firm's human capital, and the market. The optimal project should enable the firm to generate the maximum expected profit. Moreover, to restrict our attention to technological opportunity sets where safer projects offer bigger expected cost reductions, we assume that $\gamma(\alpha_i) + \alpha_i \gamma'(\alpha_i) > 0$.

Second, for each project $\alpha$, there is an optimal human capital level. Third, based on its human capital level, the firm then decides its optimal R&D level. We know that innovation as a way of knowledge creation is an activity with a basic element of uncertainty. For project $\alpha$, based on previous study [24], we also assume that the success probability of a particular project at time $t$ in firm $i$ is given by

$$\mu_i = S_i R_i \tag{2}$$

where $R_i$ is firm $i$'s R&D expenditure and $S_i$ is firm $i$'s skilled human capital that can promote innovation. Note that both $S_i$ and $R_i$ are standardized into values with range [0, 1]. The relationship of $S_i$ and $R_i$ expressed in (2) means that we implicitly assume that they are both complementary and substitute for each other.

Thus, the success probability of innovation in firm $i$ is

$$I_i = \alpha_i S_i R_i . \tag{3}$$

This means that a firm's innovation depends not only on its R&D investment $R_i$, but also on the firm's skilled human capital and whether or not it chooses the "right" project.

Next, we used backward induction to examine the firm's strategic choices at equilibrium. First, we will examine how firms make their R&D decisions. That is, a firm first takes a project and human capital level as given and chooses its optimal R&D level, $R_i$, and then based on the optimal R&D level, it chooses optimal project, $\alpha_i$, and human capital level, $S_i$. Given project $\alpha_i$ and its human capital level, $S_i$, firm $i$ will maximize its expected profit:

$$\Pi_i = \alpha_i \mu_i \pi_i^S + (1 - \alpha_i \mu_i) \pi_i^F - r R_i - w S_i, \tag{4}$$

where $\pi_i^S$ is the profit firm $i$ will obtain if its innovation is successful and $\pi_i^F$ is its profit if its innovation fails. $r$ is the cost rate for R&D, which may include the interest rate, government incentives, and subsidies for firm R&D. $w$ is the wage for skilled workers. Equation (4) states that the expected profit is the firm's expected profit after innovation minus its expenditure on innovation, R&D, and wages for skilled workers.

The firm's after-innovation profit $\pi_i^S$ and $\pi_i^F$ are determined not only by firm $i$'s innovation but also firm $j$'s ($j \neq i$; $i, j = 1, 2$.) innovation because the two firms Cournot compete with each other in the same market. Firm $j$ also may succeed or fail in innovation; thus, we will have

$$\pi_i^S = \alpha_j \mu_j \pi_i^{SS} + (1 - \alpha_j \mu_j) \pi_i^{SF}, \tag{5}$$

$$\pi_i^F = \alpha_j \mu_j \pi_i^{FS} + (1 - \alpha_j \mu_j) \pi_i^{FF}, \tag{6}$$

where $\pi_i^{SS}$ is firm $i$'s profit when both firms succeed in their innovation, $\pi_i^{SF}$ is firm $i$'s profit when firm $i$ succeeds while firm $i$ fails, $\pi_i^{FS}$ is firm $i$'s profit when firm $i$ fails while firm $j$ succeeds, and $\pi_i^{FF}$ is firm $i$'s profit when both firms fail.

We then plugged (5) and (6) into (4), and then took the derivative with respect to $R_i$, and thus, we get the reaction function of firm $i$. Following the same procedure, we obtain the reaction function of firm $j$. Combine the two reaction functions, and we then solve for the optimal R&D, $R_i$, given $\alpha_i$,

$$R_i^* = \frac{-r + 4\alpha_j \gamma(\alpha_j) S_j (A + c_i - 2c_j + \gamma(\alpha_j))}{4\alpha_i \alpha_j \gamma(\alpha_i) \gamma(\alpha_j) S_i S_j}. \tag{7}$$

Equation (7) shows that a firm's R&D spending is determined not only by its own and rival's technology level (cost function) ($c_i$, $c_j$), but also by both firms' skilled human capital level and their project choices ($S_j$, $S_j$, $\alpha_i$, $\alpha_j$).

When we took derivatives with respect to different variables and parameters, respectively, we could analyze a firm's R&D behavior more specifically. Under assumption $\gamma(\alpha_i) + \alpha_i \gamma'(\alpha_i) > 0$, we found that the derivative of $R_i^*$ with respect to $\alpha_i$ is smaller than zero, implying that the firm will choose more R&D spending if they choose a riskier project, consistent with our intuition. By taking the derivative with respect to $c_i$ and $c_j$, respectively, we found that the firm will invest more R&D if it has less advanced technology and if its rival has more advanced technology. That is, firms in a market with laggard technology have less R&D investment than those in a market with advanced technology. Similarly, we concluded that R&D is increasing with its rival's human capital and decreasing with its own human capital level. It is very easy to understand since the firm tends to invest more R&D when the competition is more intensive. Because of the substitution effect between

R&D and human capital level, more human capital tends to induce less R&D. Similarly, we found that firms in a market with higher human capital levels have more R&D investments than firms in a market with lower human capital levels.

Moreover, firms can choose their optimal human capital level. We could solve that

$$S_i^* = \frac{\sqrt{(-2A + 4c_i - 2c_j + \gamma(\alpha_j))r}}{2\sqrt{w}\sqrt{\alpha_i}\sqrt{\gamma(\alpha_i)}} \ . \tag{8}$$

We could see that a firm's skilled human capital decreases with local wage costs, and its rival's cost but increases with its own cost. In addition, it increases with R&D cost, $r$. This is the substitution effect between R&D and human capital. Similar to $R_i^*$, when under the assumption $\gamma(\alpha_i) + \alpha_i\gamma'(\alpha_i) > 0$, we found that the derivative of $S_i^*$ with respect to $\alpha_i$ is smaller than zero, implying that the firm will choose a higher human capital level if they choose a riskier project, consistent with our intuition.

Finally, we solved the optimal project. We plotted the relationship between profit and projects in Figure 1. We could see that the profit is a concave function of the project, $\alpha$, and there is an optimal value, $\alpha^*$, so that a firm can receive maximum profit.

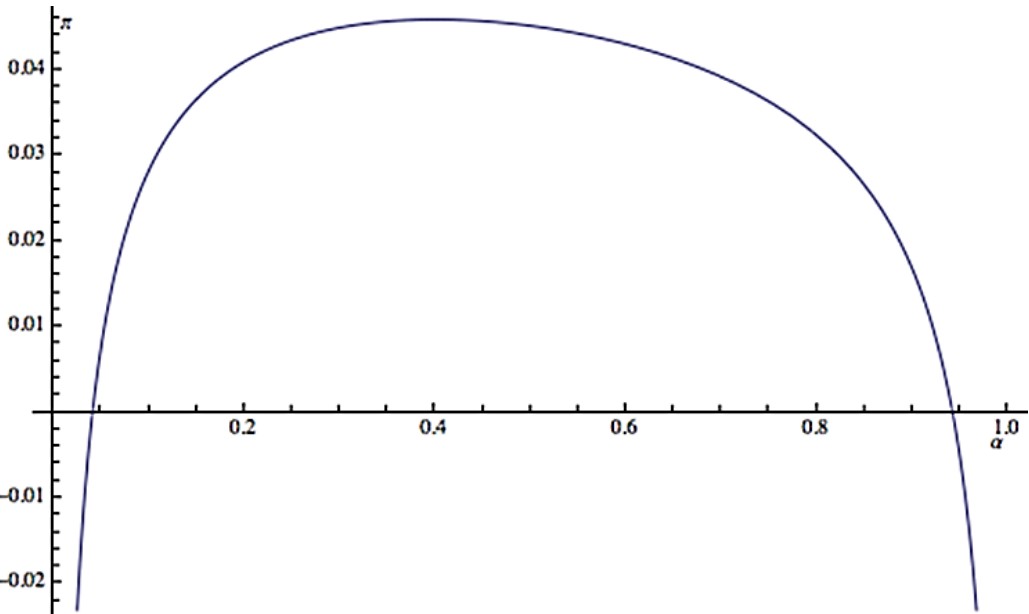

**Figure 1.** Profit and innovation project.

Plugging $R_i^*$, $S_i^*$ and $\alpha_i^*$ into (3), we finally obtained the innovation level at equilibrium

$$I_i^* = I_i(A, c_i, c_j, \ w, r) \ . \tag{9}$$

Therefore, at equilibrium, firm $i$'s innovation is determined by market demand (A), its own and its rival's technology level, and R&D cost rate and wage.

However, in reality, because of asymmetric information, imperfect decision processes, financial constraints, and some other reasons (real project choice, $\alpha_i^r$, and real R&D level), $R_i^r$, cannot be the optimal levels. That is, $\alpha_i^r \neq \alpha_i^*$ and $R_i^r \neq R_i^*$. Among all reasons, managerial human capital in a firm is usually an essential factor that affects the difference between actual decisions and optimal levels. We have

$$|\alpha_i^r - \alpha_i^*| = \varphi^\alpha(M_i), \tag{10}$$

and

$$|R_i^r - R_i^*| = \varphi^R(M_i), \tag{11}$$

where $M_i$ is a firm's managerial human capital, $\varphi^\alpha(M_i)$ is the distance between the actual project and the optimal project, and $\varphi^R(M_i)$ is the distance between the actual R&D expenditure and the optimal R&D level. Here, we use the absolute value of the difference to measure distance. Moreover, we had $\frac{\partial \varphi^\alpha(M_i)}{\partial M_i} < 0$ and $\frac{\partial \varphi^R(M_i)}{\partial M_i} < 0$, implying that the higher managerial human capital, the closer the actual decision and the optimal level.

Finally, the actual innovation level is determined by

$$I_i^r = I_i\left(A, R_i, R_j, S_i, S_j, c_i, c_j, M_i, M_j, w\right). \tag{12}$$

In Equation (12), we also included $R_i, R_j, S_i, S_j$ to control for the influential factors other than what we considered in our model. In sum, our theoretical framework indicated that a firm's innovation ($I_i^r$) is determined by a combination of the firm's skilled human capital ($S_i$), managerial human capital ($M_i$), firm R&D ($R_i$), and market demand ($A$). Thus, in our empirical study, we not only needed to include a firm's skilled human capital ($S_i$), managerial human capital ($M_i$), firm's R&D ($R_i$), firm characteristics, and market share in our estimation but also the market environment. We use firm characteristics to control for the firm's cost ($c_i$) and market share for the demand faced by the firm. Moreover, we used two datasets to control the effects of the market environment (or the other firm) on firm innovation.

## 4. Empirical Strategy

### 4.1. Specification

From Equation (12), we concluded that skilled human capital in a firm and its managerial human capital together with firm R&D all are vital for a firm's innovation. Thus, they should be included in studying firm innovation. In addition, firm characteristics, i.e., firm size, firm age and ownership structure, market structure, industry fixed effect, and city fixed effect, are also controlled. Moreover, R&D is added to the specification to control factors affecting R&D other than the variables we already controlled. In reality, innovation is usually tough to measure, and a common practice is to use the number of patent applications to measure innovation. It is assumed that

$$pat_i = \rho_i I_i^r , \tag{13}$$

where $\rho_i$ is the patenting propensity ratio of a firm, usually determined by the characteristics of innovation, firm size, government policy, and some other factors inside the firm.

Combining Equations (12) and (13), the knowledge production function in our study is specified as

$$\begin{aligned} log(pat_{it}) = \beta 0 + \beta 1 HC_{it} + \beta 2 log(RD_{it}) + \beta 3 SZ_{it} + \beta 4 MKTSHR_{it} \\ + \beta 5 W_{it} + u_{it}, \end{aligned} \tag{14}$$

where $pat_{it}$ is the number of patents applied for in China, $HC_i$ is human capital indicators, $RD_i$ is R&D expenditure, $SZ_{it}$ is firm size, $MKTSHR_{it}$ is market share, $W_{it}$ is some control variables, such as industry and city fixed effect, and $u_i$ is a disturbance term, assumed to be distributed independently but not necessarily identically across firms, for firm $i = 1, 2, \dots, n$.

The patent application was used to measure firm innovation. We used skilled human capital, the general manager (GM)'s tenure and education, and the average age and education of the management team as our human capital indicators. The GM's tenure is the years the GM holds their position. We used the number of highly educated workers or skilled workers to measure a firm's skilled human capital. We used the GM's graduate degree dummy to account for their education. The education of the management team is the average years of schooling of the management team.

However, an identification problem, ignored by almost all Schumpeterian studies, arises as we include a firm's skilled human capital level in our estimation because factors

affecting a firm's workforce adjustment are very likely to be related to factors affecting the firm's innovation. For example, a firm that wants to be active in innovation tends to hire more highly educated workers. Previous studies mentioned that by employing a larger share of skilled labor, firms can reduce informational leakages and spillovers, which outside competitors can freely acquire, and thereby lessen the threat of imitation and technological leapfrogging because of the tacit knowledge and non-codified know-how embedded in skilled workers [25]. Moreover, successful innovation may also increase the proportion of skilled workers in the workforce because more advanced technology needs to be complementary to be productive [26]. The endogeneity of skill adjustments in response to technological changes within a firm is also mentioned by the previous literature [27]. Following their method, we used the number of applicants for the positions, the average number of days those positions are vacant, the percentage of redundant unskilled workers, and city-level average wage as instruments.

Another critical variable in the patent production function is R&D spending by the firm. Contemporaneous R&D is very likely to be endogenous. That is, there is a possible correlation between unobserved innovation productivity shocks and R&D level. Thus, we exclude current R&D from the averages to lessen endogeneity.

Firm size is measured by the log of total assets rather than the log of total sales to lessen the correlation between firm size and other variables. Intuitively, firms with more resources will tend to innovate more because they have the ability to innovate. Generally, we expected a positive effect of firm size when human capital is considered.

We used two approaches to study the effect of the market environment on innovation. First, we included the city-level GDP and market share of each firm in our model. Second, we used two datasets, one from metropolitan cities and the other from provincial middle cities, to examine how firms in different markets (metropolitan cities versus mid-sized cities) innovate.

### 4.2. Regression Model of Count Data

The number of patents applied for by a firm is a count variable, so we need to use models of count data. In the following, we would introduce the Poisson model, Poisson QMLE (quasi-maximum likelihood estimator), and negative binomial model in the framework of a linear exponential family (LEF). This presentation follows the previous literature [28]. A density $f_{LEF}(y|\mu)$ is a member of a linear exponential family if

$$f_{LEF}(y|\mu) = \exp\{a(\mu) + b(y) + c(\mu)y\}, \tag{15}$$

where the function $b(.)$ is a normalizing constant, and $\mu = E[y]$, and the function $a(.)$ and $c(.)$ are such that

$$E[y] = -\left[c'(\mu)\right]^{-1} a'(\mu), \tag{16}$$

where $a'(\mu) = \frac{\partial a(\mu)}{\partial \mu}$ and $c'(\mu) = \frac{\partial c(\mu)}{\partial \mu}$, and

$$V(y) = [c\prime(\mu)]^{-1}. \tag{17}$$

Different functional forms for $a(.)$ and $c(.)$ lead to different LEF models. Special cases of the LEF include Poisson and binomial (with the number of trials fixed), and exponential. For example, the Poisson density can be written as $\exp\{-\mu + y \, ln\mu - lny!\}$, which is an LEF model with $a(\mu) = -\mu$, $c(\mu) = ln\mu$, and $b(y) = -lny!$

A regression model is formed by specifying the density to be $f_{LEF}(y_i|\mu_i)$ where $\mu_i = \mu(X_i, \beta)$, for some specified mean function $\mu(.)$. The MLE based on an LEF, $\hat{\beta}_{LEF}$ maximizes

$$L_{LEF} = \sum_{i=1}^{n} \{a(\mu_i) + b(y_i) + c(\mu_i)y_i\}. \tag{18}$$

The first-order conditions can be written as

$$\sum_{i=1}^{n} \frac{1}{v_i}(y_i - \mu_i)\frac{\partial \mu_i}{\partial \beta} = 0, \tag{19}$$

where $v_i = \left[c'(\mu_i)\right]^{-1}$ is the specified variance function that is a function of $\mu_i$ and hence $\beta$.

Under the standard assumption that the density is correctly specified, then we have

$$\sqrt{n}(\hat{\beta}_{LEF} - \beta_0) \xrightarrow{d} N\left(0, \, A^{-1}\right), \tag{20}$$

where $A = \lim\limits_{n \to \infty} \frac{1}{n} \sum\limits_{i=1}^{n} \frac{1}{v_i} \frac{\partial \mu_i}{\partial \beta} \frac{\partial \mu_i}{\partial \beta'}\bigg|_{\beta_0}$.

When the density is misspecified, the estimator is called a quasi-maximum likelihood estimator (QMLE). In application, because it is only ordinary Poisson with robust error, we still call it Poisson in our results analysis. Gourieroux, Montfort, and Trognon (GMT) show that when the mean is correctly specified, but other features of the distribution such as the variance and density are potentially misspecified, $\hat{\beta}_{QMLE} \xrightarrow{p} \beta_0$, so the MLE is still consistent for $\beta_0$ [29]. Moreover,

$$\sqrt{n}(\hat{\beta}_{QMLE} - \beta_0) \xrightarrow{d} N\left(0, \, A^{-1}BA^{-1}\right), \tag{21}$$

where A is defined above and $B = \lim\limits_{n \to \infty} \frac{1}{n} \sum\limits_{i=1}^{n} \frac{\omega_i}{v_i^2} \frac{\partial \mu_i}{\partial \beta} \frac{\partial \mu_i}{\partial \beta'}\bigg|_{\beta_0}$. Note that $v_i$ is the working variance, the variance of the specified density for $y_i$, whereas $\omega_i$ is the variance for the true dgp. Given the specification of a true variance function, $\omega_i = \omega(.)$, one can potentially obtain a more efficient estimator. The negative binomial model with mean $\mu$ and variance $\mu + \alpha\mu^2$ is one of the examples. We could see that NB model generalizes the Poisson QMLE model by allowing for an additional source of variance. For the Poisson model, the variance is restricted to equal to its mean, the so-called equi-dispersion, while the negative binomial model allows for over-dispersion.

## 5. Data

In this paper, we used data from two surveys. The first is "The study of competitiveness, technology and firm linkage" conducted by the World Bank in China in 2002. The second is "Investment climate survey" conducted also by the World Bank in 2003. Though with different names, these two surveys are very similar. Both collected information on innovation and technology, firm productivity, finance, labor, the obstacles to doing business, etc. Both are filled out by the senior manager of the main production facility of the firm and the accountant and/or personnel manager of the firm. The first dataset was carried out in 2001–2002, and covered firms in five big cities, Beijing, Chengdu, Guangzhou, Shanghai, and Tianjin. The sample includes 1548 observations and 1206 variables. Most quantitative questions covered the period 1998–2000; most qualitative questions covered only the time of the survey, 2000 (we call the first dataset Data2000, thereafter). The second dataset was conducted in 2003 and covered firms in 18 cities, smaller than those surveyed in 2000. The 18 cities are Benxi, Changchun, Changsha, Chongqing, Dalian, Guiyang, Harbin, Hangzhou, Jiangmen, Kunming, Lanzhou, Nanchang, Nanning, Shenzhen, Wenzhou, Wuhan, Xi'an, and Zhengzhou. Most quantitative questions covered the period 2000–2002; most qualitative questions covered only the year 2002 (we call the second dataset Data2002, thereafter). Both samples consist of both manufacturing and service firms. This sample includes 2400 establishments and 1073 variables.

The World Bank enterprise survey was conducted in both large firms and SMEs, and provides detailed information on human capital, finance, and innovation. Thus, the data can hardly be replaced by data from other sources. The enterprise survey series conducted by the World Bank in China include 2002, 2003, 2005, and 2012. Only the 2002

and 2003 waves used in our study are comparable. They both have the human capital information, and detailed firm finance and innovation information. In the 2005 survey, there is no firm innovation information, i.e., patent information or new product information. The 2012 survey lacks patent information, as well as the information on general manager. Therefore, we cannot do the similar analysis using more updated data. However, by focusing on firm innovation in different market environment from a human capital point of view, the paper could shed light on firm innovation for firms in other developing countries.

Firm-level survey data in China are rare, especially data with detailed information on innovation and on labor force characteristics in small and medium enterprises. The World Bank survey data are probably the most common data used in the literature [16,27,30]. Therefore, studies utilizing the rich information in this survey still provide important implications for developing countries.

The data are randomly selected from all firms in their respective cities and industries. The resulting size range is extreme, with the reported number of production workers ranging from 3 to 83,542 in Data2000 and from 1 to 70,169 in Data2002. To reduce the heterogeneity among firms, we restrict our data only to the manufacturing industry and confine our research to the subsample with at least 50 total workers, at least 10 highly educated workers and 10 less educated workers and RMB 3000,000 sales. As a result, there are 624 firms in Data2000 and 913 firms in Data2002.

We then presented a statistics summary for the full sample in Table 1. We could see that firms in Data2000 do better than firms in Data2002 with average 0.84 patents in the year 2000 in Data2000 and average 0.74 patents in the year 2002 in Data2002. Generally, firms are bigger in Data2002, and have more highly educated workers and more total workers, with around 184 highly educated workers and 945 total workers in Data2000 and around 157 highly educated workers and 736 total workers in Data2002. We can see that the sales of firms in metropolis (Data2000) are more than those in provincial big cities (Data2002), though the difference is not big, at around RMB 63 million. However, there is a large difference in R&D between the two datasets, with around RMB 19 million in Data2000 and around RMB 4 million in Data2002. Another essential difference between the two datasets is that firms in Data2000 have a higher market share (16.13%) than in Data2002 (8.96%). In addition, there is little difference in the general manager's education and experience and the firm's age.

**Table 1.** Descriptive statistics.

| Variable | Obs | Data2000 Mean | Std. Dev. | Obs | Data2002 Mean | Std. Dev. |
|---|---|---|---|---|---|---|
| Number of patents applied by firm | 624 | 0.84 | 4.28 | 910 | 0.71 | 3.68 |
| Skilled workers (hundred) | 623 | 1.84 | 3.25 | 904 | 1.57 | 2.86 |
| Total number of employees (hundred) | 624 | 9.45 | 15.05 | 909 | 7.36 | 13.21 |
| R&D (RMB thousand) | 603 | 18,996.44 | 237,062.50 | 904 | 4140.77 | 26,450.19 |
| Value of total sales (RMB million) | 624 | 334.31 | 1828.58 | 909 | 271.05 | 1246.51 |
| Total net assets (RMB million) | 622 | 102.79 | 430.07 | 905 | 96.52 | 411.77 |
| Years of schooling of GM | 622 | 14.03 | 2.30 | 903 | 14.15 | 2.23 |
| Years of GM holding the position | 623 | 5.69 | 4.44 | 901 | 5.86 | 4.47 |
| GM's postgraduate | 622 | 0.16 | 0.37 | 903 | 0.17 | 0.37 |
| Management team's average age | 614 | 36.29 | 6.63 | 879 | 36.50 | 5.31 |
| Management team's average schooling | 615 | 11.88 | 1.50 | 883 | 12.13 | 1.50 |

**Table 1.** *Cont.*

| Variable | Obs | Data2000 Mean | Std. Dev. | Obs | Data2002 Mean | Std. Dev. |
|---|---|---|---|---|---|---|
| Firm's market share | 583 | 16.13 | 20.53 | 884 | 8.96 | 16.38 |
| Firm age | 624 | 17.81 | 17.37 | 910 | 15.96 | 14.34 |
| Shareholding firms dummy | 624 | 0.16 | 0.37 | 910 | 0.29 | 0.45 |
| State-owned firms dummy | 624 | 0.24 | 0.43 | 910 | 0.26 | 0.44 |
| Foreign invested firms dummy | 624 | 0.39 | 0.49 | 910 | 0.22 | 0.41 |

## 6. Results and Discussion

Table 2 reports the results from regressing the number of patent applications on human capital and the market environment using three different regression specifications: OLS, Poisson, and negative binomial. Columns (1)–(3) in part I are estimated using only cross-sectional data in the year 2000 in Data2000, while in part II cross-sectional data in the year 2002 in Data2002 are used because our human capital indicators are only available for the survey year. All specifications include ownership dummies, city dummies, and industry dummies.

**Table 2.** Baseline results.

| | I: Year 2000 (Data 2000) | | | II: Year 2002 (Data 2002) | | |
|---|---|---|---|---|---|---|
| | OLS | Poisson | NB | OLS | Poisson | NB |
| | (1) | (2) | (3) | (4) | (5) | (6) |
| Skilled workers (hundred) | 0.425 *** | 0.0831 *** | 0.292 *** | 0.536 * | 0.0855 *** | 0.164 *** |
| | (0.162) | (0.0172) | (0.0766) | (0.319) | (0.0201) | (0.0468) |
| GM's experience (years) | 0.201 ** | 0.200 *** | 0.171 *** | 0.0161 | 0.00515 | 0.0112 |
| | (0.0856) | (0.0328) | (0.0404) | (0.0225) | (0.0307) | (0.0299) |
| GM's postgraduate degree | 1.004 * | 0.638 | 0.353 | 0.633 | 0.620 * | 0.835 *** |
| | (0.609) | (0.398) | (0.365) | (0.440) | (0.332) | (0.283) |
| Management team's average age | −0.0552 | −0.0397 | −0.0853 *** | −0.00273 | −0.0615 ** | −0.0758 *** |
| | (0.0392) | (0.0327) | (0.0290) | (0.0230) | (0.0264) | (0.0250) |
| Management team's average schooling | 0.0370 | 0.114 | 0.354 *** | 0.0981 | 0.103 | 0.0649 |
| | (0.0749) | (0.0897) | (0.136) | (0.0841) | (0.0999) | (0.0960) |
| Market share | 0.0156 | 0.00899 | 0.0242 *** | 0.0124 | 0.0154 *** | 0.0294 *** |
| | (0.0102) | (0.00640) | (0.00754) | (0.00775) | (0.00527) | (0.00556) |
| Log (R&D) | −0.00874 | −0.0142 | −0.0211 | 0.0294 * | 0.0963 *** | 0.0958 *** |
| | (0.0317) | (0.0311) | (0.0282) | (0.0162) | (0.0245) | (0.0217) |
| Firm size (log (total net assets)) | 0.0355 | 0.346 *** | 0.0872 | −0.122 | 0.145 | 0.0512 |
| | (0.141) | (0.127) | (0.137) | (0.186) | (0.105) | (0.0947) |
| Firm age (year) | −0.00278 | −0.00532 | 0.00505 | −0.0149 | −0.0137 | −0.00999 |
| | (0.00840) | (0.0126) | (0.0113) | (0.00952) | (0.0133) | (0.0112) |
| Log (GDP per capita) | 4.988 * | 3.454 | 0.897 | 2.986 | −0.740 *** | −0.666 * |
| | (2.574) | (3.065) | (3.649) | (4.191) | (0.220) | (0.364) |
| Constant | −49.85 ** | −40.93 | −15.28 | −28.49 | 4.274 * | 5.631 |
| | (24.61) | (30.11) | (36.07) | (39.79) | (2.499) | (3.811) |
| Number of observations | 562 | 562 | 562 | 824 | 824 | 824 |

Note: (1) Robust standard errors in parentheses: * $p < 0.10$, ** $p < 0.05$, *** $p < 0.01$. (2) In the negative binomial (NB) model, the coefficient of log(alpha) is 2.350 with the standard error 0.178, indicating the existence of overdispersion. (3) Ownership, year dummies, and industry dummies are also controlled.

OLS estimator is the simplest to use and requires the least requirements to be consistent, but it ignores the count nature of the data. Poisson with robust errors and negative binomial estimates take both count data nature and overdispersion into account. To obtain consistent estimates, the Poisson model only requires that the conditional mean is correctly

specified, while negative binomial estimates require not only the correctly specified mean condition but also the variance condition. The fact that we need to correctly specify the mean condition as well as the variance condition results in a stronger assumption. The stronger assumption can lead to more efficient estimation. However, if the additional variance condition is not correctly specified, the estimates become inconsistent. The negative binomial model fits the data better, and thus our analysis will be based on it. Moreover, the parameter of overdispersion is significant in both datasets, supporting the negative binomial model.

The first significant result is that the number of highly educated workers has a positive and significant coefficient across all models and both datasets, suggesting a positive effect of skilled human capital on innovation. That is, when a firm has more skilled human capital, it will tend to have more innovation. Specifically, we obtained a coefficient of 0.292 using the year 2000 data and a coefficient of 0.164 using the year 2002 data and both are significant at 1% level, indicating that if other things are equal, when skilled workers increase by 100 people, the number of patents will increase by 29.2 % in Data2000 and 16.4% in Data2002, respectively. When evaluating at sample average, we found that when skilled workers increase by 1%, the number of patents will increase by 0.54% in Data2000 and 0.26% in Data2003, respectively. The broad conclusions we drew from these estimates are similar in spirit to Bosetti et al. and Laursen et al. [31,32], who found that highly educated workers positively affect firm innovation. Moreover, Andersson and Lööf found that when skilled labor increases by 1% [33], the number of patent application will increase by 0.191–0.6% in the negative binomial model. Thus, the results are comparable.

We could see that the effect of skilled human capital is quite significant both statistically and economically, and it is robust across both datasets. Their results indicated that, if other things are equal, when the number of skilled labor increases, firm innovation would also increase. Very interestingly, we noticed that the effect of skilled human capital has a larger effect in Data2000 than in Data2002, even when there is a time trend in Data2002, implying how the market environment might influence the effect of skilled human capital on innovation. Skilled workers are more important for firm innovation in metropolitan cities.

The GM's experience is positive and significant both statistically and economically in Data2000. In the NB model, we found the coefficient is around 0.171, which means that when the GM holds the position for one additional year, the number of patent applications will increase by 17.1%. This is consistent with the literature that showed that the GM's tenure has a positive effect on R&D expenditure [16]. They argued that shorter-tenured GMs might have greater incentives to focus on short-term outcomes in order to build their reputation, and therefore might be less willing to invest in high-risk long-horizon R&D projects. Wu et al. showed that the GM's tenure has a positive but insignificant effect on R&D expenditure [34]. We found that a positive effect of the GM's tenure might be that a GM with longer tenure can be more experienced with the firm and the market structure and technology opportunity in this industry, and can thus have a good judgment regarding a firm's innovative capacity and market demand. This is especially true for firm innovation that is full of uncertainty. However, some studies found that general managers tend to make fewer changes in strategy as their tenure increases. Islam and Zein found that CEO tenure tends to have a negative effect on patents [35]. The reason might be that CEO tenure is much longer in their datasets (7.82 years for non-inventor CEOs and 12.03 for inventor CEOs). Some scholars claimed that this lack of change occurs because when tenure increases, the GM becomes conservative and more strongly committed to implementing their own paradigm for how the organization should be run [36]. The positive effect of GM tenure in our study indicated that the effect of good judgment is larger than the effect of conservative leadership.

We included the GM's postgraduate degree to indicate the GM's education because there are more than 70% of firms' GMs with a college degree, and thus, under this situation, the study of a postgraduate degree will be more meaningful. Table 2 shows that the GM's graduate degree is insignificant in NB model in Data2000, while it is significant both in

the Poisson model and the NB model in Data2002. In the NB model, the coefficient of a postgraduate degree is 0.835, indicating that when a firm hires a GM with a postgraduate degree, its innovation will increase by 83.5% compared to firms having a GM without graduate degrees in mid-sized cities.

Moreover, Table 2 shows that the management team's average age has a negative and significant coefficient in both datasets, while their average schooling tends to have a positive coefficient only in metropolitan cities. Thus, we concluded that the management team's average age has a negative effect on innovation while their average education tends to positively affect innovation. Specifically, other things equal, when the management team's average age increases by one year, the firm's number of patents will decrease by 8.53% in metropolitan cities and decrease by 7.58% in mid-sized cities. Our results are consistent with our intuition and previous management studies. Older executives tend to be more conservative [20], and empirical studies have found that older top managers tend to be risk averse [37] and follow lower-growth strategies. One reason is that older executives have less of the physical and mental stamina needed to implement organizational changes. Another reason is that older managers may have greater difficulty grasping new ideas and learning new behaviors [20] because some cognitive abilities seem to diminish with age, including learning ability, reasoning, and memory. Finally, younger managers are likely to have received their education more recently than older managers, so their technical knowledge should be superior.

Meanwhile, Table 2 shows that the management team's average schooling has a positive and significant effect in metropolitan cities while insignificant in mid-sized cities. The importance of the top manager's education has been studied in a number of studies. Attained education level is always assumed to be correlated with cognitive ability. Moreover, higher levels of education are often assumed to be associated with higher ability to generate (and implement) creative solutions to complex problems.

R&D is insignificant in metropolitan cities while it is significant and positive in mid-sized cities. In mid-sized cities, when R&D increases by 10%, if other things are equal, the number of patents will increase around by 1%. We claim that the relationship between R&D and innovation, when measured by patents, may be affected by market environment. In the theoretical part, we have shown that the market environment with advanced technology and more human capital stimulates firm R&D. Moreover, in the datasets, R&D in Data2000 is around five times R&D in Data2002. Thus, we concluded that it might be that firms in metropolitan cities might have enough R&D, so the variation in R&D is not the reason why our dependent variable, the number of patents, varies.

Market share has a positive effect across all models and is significant in Poisson in Part I and significant in Poisson and negative binomial in Part II, strongly supporting Schumpeterian hypotheses. This is not hard to understand. With a bigger market share, firms can have more profit, and thus firms can have more resources to put into R&D. This is important because possible failures in financial markets may force firms to rely on their own supra-normal profits to finance the search for innovation. Moreover, with a bigger market share, firms can appropriate more profits from more sales using innovation.

GDP per capita tends to have a positive effect in metropolitan cities, although insignificant, and it has a negative and significant effect in mid-sized cities. It is not hard to understand the positive relationship between GDP per capita and firm innovation, since economic incentives foster innovation. Firms located in cities with higher per capita are more likely to invest in R&D [38,39].

The negative relationship between GDP per capita and firm innovation is somewhat surprising. However, this is consistent with the literature. Using Eurostat and national-regional data from all European Union countries, scholars found that in Germany and the UK, GDP has a strong positive influence on firm innovation, but the influence is negative in France [40]. They found that industry composition could account for their results. Germany and the UK are two traditional industrial nations. Firms in these two countries contribute to GDP as well as innovation efforts. However, the French economy is dominated by the

service sector, where patents are much less often used to protect intellectual property than in the industrial sector.

Following their logic, the industry composition explanation could also account for our results. Figure 2 shows the average patents over industries in mid-sized cities. We could see that firms in the household electronics industry have the most average patent application, 1.93, followed by the firms in the auto and auto parts industry. Meanwhile, firms in garment and leather products, transportation equipment, and chemical product and medicine have the least patents, which are around 0.1. Therefore, there are large variances in patent applications across different industries. Table 3 presents the number of firms in different industries over cities. Shenzhen ranks number one in GDP per capita, but its firms mainly lie in electronic equipment, garment and leather products, and electronic parts making industries. On the other hand, Chongqing has the lowest GDP per capita, but its firms mainly lie in patents intensive industries, auto and auto parts. Therefore, the industry composition is the main reason for the negative relationship between GDP per capita and firm innovation in mid-sized cities.

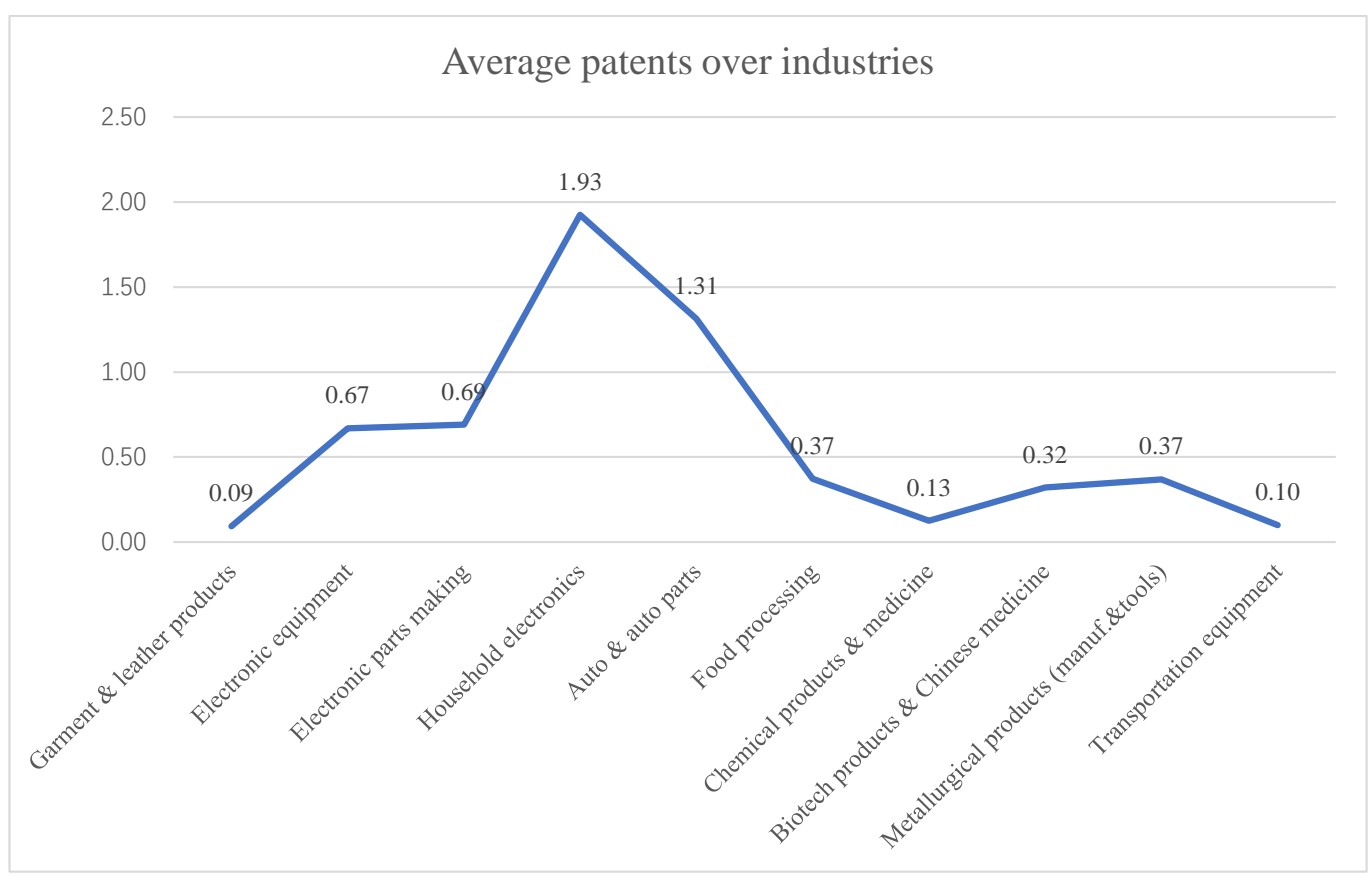

**Figure 2.** Average patents over industries (Data2002).

**Table 3.** Number of firms in different industries in cities.

| | GDP per Capita | (1) | (2) | (3) | (4) | (5) | (6) | (7) | (8) | (9) | (10) | Total |
|---|---|---|---|---|---|---|---|---|---|---|---|---|
| Benxi | 12,362 | 2 | 0 | 0 | 0 | 0 | 4 | 2 | 2 | 12 | 3 | 25 |
| Changchun | 16,220 | 5 | 4 | 3 | 0 | 33 | 2 | 4 | 1 | 20 | 0 | 72 |
| Changsha | 13,747 | 14 | 5 | 14 | 1 | 25 | 0 | 0 | 0 | 0 | 0 | 59 |
| Chongqing | 6347 | 12 | 8 | 13 | 2 | 39 | 0 | 0 | 0 | 0 | 0 | 74 |
| Dalian | 25,276 | 13 | 9 | 12 | 2 | 10 | 3 | 0 | 0 | 1 | 0 | 50 |
| Guiyang | 9948 | 4 | 5 | 12 | 0 | 10 | 8 | 6 | 12 | 0 | 1 | 58 |
| Haerbin | 12,993 | 4 | 3 | 9 | 1 | 11 | 0 | 0 | 0 | 0 | 0 | 28 |
| Hangzhou | 28,150 | 13 | 12 | 15 | 10 | 13 | 0 | 0 | 0 | 0 | 0 | 63 |
| Jiangmen | 17,344 | 22 | 1 | 10 | 9 | 8 | 0 | 0 | 0 | 0 | 0 | 50 |
| Kunming | 14,864 | 5 | 0 | 7 | 0 | 7 | 0 | 25 | 0 | 0 | 0 | 44 |
| Lanzhou | 12,588 | 3 | 0 | 0 | 0 | 0 | 2 | 0 | 0 | 24 | 0 | 29 |
| Nanchang | 12,552 | 16 | 3 | 5 | 0 | 17 | 15 | 0 | 0 | 5 | 1 | 62 |
| Nanning | 12,024 | 4 | 0 | 0 | 0 | 4 | 9 | 3 | 10 | 1 | 5 | 36 |
| Shenzhen | 46,388 | 14 | 22 | 13 | 5 | 6 | 0 | 0 | 0 | 0 | 0 | 60 |
| Wenzhou | 14,357 | 11 | 4 | 12 | 10 | 7 | 0 | 0 | 0 | 2 | 0 | 46 |
| Wuhan | 19,792 | 19 | 21 | 13 | 0 | 17 | 0 | 0 | 0 | 0 | 0 | 70 |
| Xian | 11,786 | 7 | 8 | 25 | 0 | 12 | 0 | 0 | 0 | 0 | 0 | 52 |
| Zhengzhou | 14,414 | 15 | 4 | 5 | 0 | 11 | 0 | 0 | 0 | 0 | 0 | 35 |
| Total | | 183 | 109 | 168 | 40 | 230 | 43 | 40 | 25 | 65 | 10 | 913 |

Note: (1) garment and leather products industry; (2) electronic equipment industry; (3) electronic parts making industry; (4) household electronics industry; (5) auto and auto parts industry; (6) food processing industry; (7) chemical products and medicine industry; (8) biotech products and Chinese medicine industry; (9) metallurgical products industry; (10) transportation equipment industry.

Meanwhile, metropolitan cities often have a much broader spectrum of industries, and the industries are more likely to be high-tech, and therefore, their new knowledge could also be more likely to be patented. Thus, metropolitan cities tend to show a positive influence of GDP on firm innovation.

To further study the effect of the market environment, we presented the effects of the interaction of human capital and the market environment on firm innovation in Table 4. All specifications are estimated using NB. Results showed that the effect of the GM's experience is increasing with GDP. That is, in metropolitan cities with higher GDP, the firms with a more experienced GM will be rewarded more in terms of innovation. Another important finding is that the effect of R&D decreases with GDP, implying that when a mid-sized city has better economic development, the effect of R&D on firm innovation decreases. This is consistent with our main finding in Table 2 that R&D has a positive and significant effect in mid-sized cities while it is insignificant in metropolitan cities.

**Table 4.** Interaction of human capital and market environment in firm innovation.

| | I: Year 2000 (Data 2000) | | | II: Year 2002 (Data 2002) | | |
|---|---|---|---|---|---|---|
| | (1) | (2) | (3) | (4) | (5) | (6) |
| Skilled workers (hundred) | 0.810 | 0.279 *** | 0.296 *** | −1.085 | 0.164 *** | 0.160 *** |
| | (1.547) | (0.0744) | (0.0803) | (0.762) | (0.0474) | (0.0491) |
| GM's experience (years) | 0.171 *** | −1.631 ** | 0.173 *** | 0.0119 | -0.411 | 0.0199 |
| | (0.0395) | (0.711) | (0.0383) | (0.0296) | (0.697) | (0.0283) |
| GM's postgraduate | 0.334 | 0.404 | 0.298 | 0.908 *** | 0.838 *** | 0.971 *** |
| | (0.377) | (0.361) | (0.380) | (0.275) | (0.285) | (0.276) |
| Management team's average age | −0.0861 *** | −0.0764 *** | −0.0862 *** | −0.0800 *** | −0.0743 *** | −0.0768 *** |
| | (0.0288) | (0.0283) | (0.0278) | (0.0245) | (0.0246) | (0.0248) |
| Management team's average schooling | 0.349 ** | 0.344 ** | 0.347 ** | 0.0657 | 0.0496 | 0.0682 |
| | (0.141) | (0.138) | (0.136) | (0.0958) | (0.0871) | (0.0916) |
| Skilled workers × Log (GDP per capita) | −0.0510 | | | 0.135 | | |
| | (0.148) | | | (0.0850) | | |
| GM's experience × Log (GDP per capita) | | 0.178 ** | | | 0.0429 | |
| | | (0.0704) | | | (0.0708) | |
| Log (R&D) × Log (GDP per capita) | | | 0.0275 | | | −0.0797 * |
| | | | (0.0578) | | | (0.0430) |

**Table 4.** *Cont.*

| | I: Year 2000 (Data 2000) | | | II: Year 2002 (Data 2002) | | |
| --- | --- | --- | --- | --- | --- | --- |
| | **(1)** | **(2)** | **(3)** | **(4)** | **(5)** | **(6)** |
| Market share | 0.0242 *** | 0.0222 *** | 0.0240 *** | 0.0294 *** | 0.0286 *** | 0.0290 *** |
| | (0.00727) | (0.00713) | (0.00738) | (0.00558) | (0.00563) | (0.00553) |
| Log (R&D) | −0.0163 | −0.0150 | −0.294 | 0.0962 *** | 0.0965 *** | 0.871 ** |
| | (0.0278) | (0.0280) | (0.586) | (0.0212) | (0.0221) | (0.419) |
| Firm size (log (total net assets)) | 0.0659 | 0.0592 | 0.0469 | 0.00160 | 0.0552 | 0.0406 |
| | (0.143) | (0.145) | (0.141) | (0.0949) | (0.0925) | (0.0927) |
| Firm age (year) | 0.00247 | 0.00590 | 0.00328 | −0.00981 | −0.0113 | −0.00819 |
| | (0.0113) | (0.0108) | (0.0111) | (0.0110) | (0.0111) | (0.0114) |
| Log (GDP per capita) | 1.233 ** | −0.0627 | 1.091 *** | −0.894 ** | −0.934 * | −0.589 * |
| | (0.564) | (0.587) | (0.407) | (0.406) | (0.552) | (0.345) |
| Number of observations | 562 | 562 | 562 | 824 | 824 | 824 |

Note: (1) Robust standard errors in parentheses: * $p < 0.10$, ** $p < 0.05$, *** $p < 0.01$. (2) Ownership, year dummies, and industry dummies are also controlled. (3) All specifications are estimated using NB.

## 7. Further Investigation

In estimating our equation, we faced a possible econometric problem concerning the potential correlation between the independent variables, skilled human capital and R&D, and unobservable or unmeasurable firm-specific characteristics, such as the quality of human capital. The ordinary Poisson and NB estimates would then be subject to omitted-variable misspecification and bias. One of the traditional ways to correct the bias is to use panel data. With the panel data, we could demean the variables, and thus, all time-invariant firm-specific characteristics would be removed. If none of the unobservable or unmeasurable firm-specific characteristics change over time, we would obtain unbiased estimates. However, for our data, a three-year panel data, most variation of the data is cross-sectional. Applying the demean method will then wipe out the useful interfirm variation. Thus, in our study, we used cross-sectional data that make the best use of information on firm characteristics.

We used the control function approach to deal with endogeneity. Let $y_1$ denote the response variable, $y_2$ the endogenous explanatory variable, and $z$ the $1 \times L$ vector of exogenous variables (which includes unity as its first element). Consider the model

$$E(y_1|z, y_2, r_1) = \exp(z_1\delta_1 + \alpha_1 y_2 + r_1),\tag{22}$$

where $z_1$ is a $1 \times L_1$ strict subvector of $z$ that also includes a constant, and $r_1$ is the error term. Suppose first that $y_2$ has a standard linear reduced form with an additive and independent error

$$y_2 = z\pi_2 + v_2,\tag{23}$$

$$D(r_1, v_2|z) = D(r_1, v_2),\tag{24}$$

so that $(r_1, v_2)$ is independent of $z$. Then

$$E(y_1|z, y_2) = E(y_1|z, v_2) = E(\exp(r_1)|v_2)\exp(z_1\delta_1 + \alpha_1 y_2).\tag{25}$$

If $(r_1, v_2)$ are jointly normal, then $E(\exp(r_1)|v_2) = \exp(\theta_1 v_2)$, where we set the intercept to zero, assuming $z_1$ includes an intercept. This assumption can hold more generally, too. Then

$$E(y_1|z, y_2) = E(y_1|z, v_2) = \exp(z_1\delta_1 + \alpha_1 y_2 + \theta_1 v_2).\tag{26}$$

This expectation immediately suggests a two-step estimation procedure. The first step is to estimate the reduced form for $y_2$ and obtain the residuals. Second, including $\hat{v}_2$, along with $z_1$ and $y_2$, in negative binomial.

Although in the linear model, the control function estimates are identical to the 2SLS estimates, in the exponential model, we could obtain a more efficient estimator via the control function method. Moreover, we could still take the count data feature and

overidentification feature in the second stage of the control function by using the negative binomial model.

We used the number of job applicants for skilled worker positions, the number of weeks to fill the last job, and the city-level average wage as instruments for skilled human capital in Data2000. However, we had no information on job applicants and vacant positions. In Data2002, we used the percentage of redundancy in unskilled workers and city-level average wage as instruments for skilled human capital.

Table 5 shows the results of IV estimation using negative binomial control function approach. The results of the first stage regression suggest that we are using reasonably "strong" instruments. We also partially tested the validity of the instruments by the overidentification test and did not reject the null that the over-identifying instruments are valid, assuming a subset of the instruments is valid, and identified the model. The IV results in Table 5 are generally consistent with those in Table 3. That is, skilled human capital is vital for firm innovation in metropolitan cities, while R&D plays an important role in firm innovation in mid-sized cities. The GM's experience is more critical for firms in metropolitan cities, while the GM's education is more important for firms in mid-sized cities. However, the residuals from the first stage in Poisson control function and negative binomial control function are not significant. This might indicate that the endogeneity of skilled workers and R&D are rejected.

**Table 5.** IV estimation Results.

|  | I: Data2000 | | II: Data2002 | |
|---|---|---|---|---|
|  | First Stage (1) | NB CF (2) | First Stage (3) | NB CF (4) |
| Applicants of skilled position | 0.735 *** | | | |
|  | (0.122) | | | |
| Weeks to fill skilled vacant position | 0.0377 | | | |
|  | (0.0303) | | | |
| Redundant unskilled workers | | | −0.0135 ** | |
|  | | | (0.00650) | |
| Log (city-level average wage) | 1.675 ** | | 3.073 | |
|  | (0.727) | | (7.328) | |
| Skilled workers (hundred) | | 0.360 * | | 0.150 |
|  | | (0.207) | | (0.269) |
| GM's experience (years) | −0.0659 | 0.184 *** | −0.00552 | 0.0149 |
|  | (0.0446) | (0.0557) | (0.0207) | (0.0302) |
| GM's postgraduate degree | 0.0849 | 0.554 | 0.276 | 0.838 *** |
|  | (0.437) | (0.373) | (0.236) | (0.301) |
| Management team's average age | 0.0507 * | −0.0704 ** | 0.0365 * | −0.0869 *** |
|  | (0.0290) | (0.0300) | (0.0188) | (0.0285) |
| Management team's average schooling | 0.124 | 0.0359 | −0.0595 | 0.0257 |
|  | (0.130) | (0.147) | (0.0628) | (0.0960) |
| Market share | −0.00650 | 0.0206 *** | −0.000971 | 0.0313 *** |
|  | (0.00766) | (0.00708) | (0.00538) | (0.00561) |
| Log (R&D) | 0.0276 | −0.00668 | 0.0652 *** | 0.114 *** |
|  | (0.0262) | (0.0292) | (0.0142) | (0.0239) |
| Firm size (log (total net assets)) | 0.984 *** | −0.144 | 0.840 *** | 0.0525 |
|  | (0.104) | (0.256) | (0.0556) | (0.241) |
| Firm age (year) | −0.00584 | −0.0109 | −0.00977 | −0.00536 |
|  | (0.0143) | (0.0180) | (0.00738) | (0.0125) |
| Number of observations | 358 | 358 | 818 | 818 |

Note: (1) Robust standard errors in parentheses: * $p < 0.10$, ** $p < 0.05$, *** $p < 0.01$; (2) In this model, skilled human capital (number of highly educated workers) are treated as endogenous. We use the number of job applicants for skilled worker positions, the number of weeks to fill the last job, and city-level average wage as instruments for skilled human capital in Data2000. However, we have no information on job applicants and vacant positions. In Data2002, we use the percentage of redundancy in unskilled workers and city-level average wage as instruments for skilled human capital. (3) Ownership, year dummies, and industry dummies are also controlled.

We could see that our main results still hold. One important implication is that R&D might not be enough in less developed areas while it is enough and might be too much in developed areas. To promote innovation, policymakers need to improve human capital in all firms and allocate more R&D in less developed areas. Moreover, it might be more efficient to evaluate whether there is a waste of R&D in more developed areas.

## 8. Conclusions

There is very little micro-based literature on the relationship between human capital and firm innovation in developing countries. To this end, this paper contributes by presenting an in-depth analysis of the effects of human capital on firm innovation in the context of a developing economy. The findings reveal some interesting insights into innovation behavior at the firm level that have the potential to serve as inputs to policy making at different levels.

Our major findings are as follows. Skilled human capital is vital for firm innovation in metropolitan cities, while R&D plays an essential role in firm innovation in mid-sized cities. The GM's experience is more important for firms in metropolitan cities, while the GM's education is more critical for firms in mid-sized cities. The management team's education tends to have a positive effect on firm innovation, while the team's average age has a negative and significant effect on firm innovation. Moreover, GDP per capita has a positive effect on firm innovation in metropolitan cities, while it hurts firm innovation in mid-sized cities. The reason behind this is the industry composition of a city.

Our results have important implications. First, human capital, skilled human capital, and characteristics of managerial personnel are essential in determining a firm's innovation. Without considering them, the study of firm innovation may be biased because of heterogeneity. Moreover, when both variables are included, human capital can account for the impact of other innovation, i.e., all the other "on the job learning" or "learning by doing". Second, controlling market share, market fixed effect, and GDP per capita is not enough for firm innovation studies. Comparing firm innovation in different market environments is essential to study how the market environment affects a firm's innovation. Thus, in addition to knowledge spillover, we find that the strategic choice of a firm plays a vital role in how the market environment affects the firm's innovation. Innovation is multifaceted, and currently, we use patent applications to measure firm innovation. In future research, we could use product innovation and total factor productivity to examine the relationship among human capital, market environment, and firm innovation. Moreover, such a study could be conducted in firms of different size, i.e., big firms and small and medium firms (SMEs). In addition, other mechanisms, e.g., R&D cooperation, which is often considered in the literature [41], also should be examined in future research.

**Funding:** This research was funded by the National Natural Science Foundation of China (Grant #72103167).

**Institutional Review Board Statement:** Not applicable.

**Informed Consent Statement:** Not applicable.

**Data Availability Statement:** The data used to support the findings of the study are freely available at https://microdata.worldbank.org (accessed on 5 January 2021).

**Conflicts of Interest:** The authors declare no conflict of interest.

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
