# Peer review of "Human Capital, Market Environment, and Firm Innovation in Chinese Manufacturing Firms"

_sustainability, doi:10.3390/su141912642_

Round 1
Reviewer 1 Report
1) Row 80. Please, define what you mean by “highly educated workers”? Ph.D.? or Master?
2) Rows 112-113. In the sentence “Among all the resources in a firm, human capital has long been argued as a critical resource,” I suggest the author include some references to reinforce this idea.
3) Rows 114-115. I suggest the author avoid general arguments, such as “because some capabilities are based on firm-specific knowledge, and others may only be valuable.” Which capabilities are based on firm-specific knowledge? Please, specify them, and try to avoid this type of sentence in the paper.
4) Row 127. When the endogenous growth theory is cited, some references should be included.
5) Row 204. Please, explain why Si and Ri are complementary and substitutes for each other, as the author affirms, because if Si and Ri are complementary, both continue in time, but if Si substitutes Ri (for example), only one remains.
6) Rows 233-234. The author has written: “and because of the complementary effect between R&D and human capital level, more human capital tends to induce less R&D.” If R&D and human capital are complementary, should they not go in the same direction?, so more human capital tends to induce HIGHER R&D?
7) Row 255. I suggest the author erase the sentence: “Because it is too complicated to be presented here.”
8) Row 276. I suggest the authors include the variables shown in equation 12 in the paragraph (rows 276-285) to make it more straightforward. For example, “we get the real innovation level (Iri) is determined by…
9) Row 374. The data used in the paper is very old (from 20 years ago!). China has changed significantly compared to today, especially in Table 3. I suggest the author explain why he/she has chosen these old databases or use more updated data.
Author Response
Dear reviewer:
I want to express my gratitude for your thoughtful suggestions and most useful comments, which I have made every attempt to take into account in a revised version of the paper. I also have carefully addressed all comments in the attached file.
Let me again thank you for your comments and suggestions which helped me to improve the paper.
Best,

Reviewer 2 Report
Thank you for elaborating a study on the importance of external environment for firm innovation. The paper has original idea and insightful empirical findings. However, the paper lacks stong positioning with regard to previous studies.
What is the main difference of your elaborated theoretical model with the previous ones? You have shown that optimal HC is determined by external environment, but you didn't use this fact in empirical study. In empirical study you justified that innovation level is determined by external environment and HC interacts with metrics of external environment.
Second concern. By describing formula 1, you wrote that market demand and rival's const reflect external environment. Later you use the word "market share" instead of market demand. Please, check the use of these words. Also it is not clear how you pass from theoretical presentation of external environment to empirical measurements. What is external environment in your case?
Third concern. You discussed three stages of innovation in theoretical part. It is not clear how you use it or not in empirical part.
Fourth concern. By describing datasets you didn't explain why you used two of them. The main focus of your study is spatial differences of China and you use this unique feature for external environment investigation. But it is not clear from your description.
The last but not least issue. By describing your empirical results you didn't provide comparison with the previous studies. Does your empirical knowledge contribute to the existence knowledge in innovation management field?
Author Response

(The authors gave the same response as above.)

Reviewer 3 Report
Dear authors,
It seems to me that this work proposes an interesting theoretical framework that contrasts and complements the empirical work carried out. It is a relevant proposal that provides a novel and necessary view and analysis in this field.
I would include more references.
Kind regards
Author Response

(The authors gave the same response as above.)

Reviewer 4 Report
Here are my comments on the paper, Human Capital, Market Environment, and Firm Innovation in Chinese Manufacturing Firms, submitted to Sustainability.
1. The existing literature section is merely an annotated bibliography with no purpose. No attempt is made to tell the reader the "so what?" The authors should provide a summary of this section.
2. The literature review should be written in the past tense as the work has already been completed, similar with the empirical results.
3. Line 12 replace the empirical with this
4. Line 38, dollars should not be capitalized.
5. Line 92 estimates should be replaced with variables
6. Line 97 are able should be replaced with could
7. Line 120 should be becomes natural
8. Line 170, ith should be ith
9. Line 288 it should be we obtained
10. How did the authors combine equations 12 and 13?
11. Lines 339-340 the following sentence is not clear and it should be revised: to examine how firms in different markets, a more advanced one and a less advanced one, 339 innovate
12. MLE is not defined anywhere in the text. The authors should provide what MLE is for the readers. I may know it is the maximum likelihood estimator, but the authors need to clarify it for the readers.
13. Lines 364-367, are you referring to the Poisson distribution? If so, make it more clear.
14. How does the Poisson and the negative binomial differ? The authors need to make it more clear for the readers.
15. These two datasets seem useful but appear to be old. Given the age of these datasets do you think there would change and the composition of the data would be different today?
16. Lines 397-398 What are the implications for developing countries?
17. In a Poisson distribution, we would assume that the mean and the variance would be the same. So the extra variation in the data could be greater than the mean so we would have overdispersion. Thus, we would need to use the negative binomial. But if one runs the regression using the Poisson and biased standard errors. Did the authors check for this?
18. In the tables showing the empirical results, are the standard errors robust? Clustered? It is not quite clear from these tables.
19. Lines 443-444 are not clear. The authors need to clarify this sentence.
20. Line 448 after we, it should be obtained.
21. Line 500-501 is an odd sentence. This sentence should be rewritten
22. The authors used "get" throughout the paper. It is an annoying word and should be replaced.
23. Table 3, the columns should have the heading "Industry"
24. The references do not conform to the style guide of Sustainability. The authors need correct the references.
Author Response

(The authors gave the same response as above.)

Round 2
Reviewer 1 Report
Thank you for your responses to improve the paper. Congratulations on your high-quality work.
Author Response
Dear Reviewer,
We appreciate you for your precious time in reviewing our paper and providing valuable comments. It was your valuable and insightful comments that led to possible improvements in the current version. We welcome further constructive comments if any.
Thank you very much.
Best,
Xiuli Sun
Associate Professor of Economics
School of Statistics,Southwestern University of Finance and Economics
Reviewer 2 Report
The paper has been improved.
The only concern now is connected with the empirical data which covers the period twenty years ago. May be in the conclusion the author could justify, why the findings are relevant for nowdays.
Author Response
Dear Reviewer:
I appreciate you for your precious time in reviewing my paper and providing valuable comments. It was your valuable and insightful comments that led to possible improvements in the current version. I want to express my gratitude for your further thoughtful suggestion and most useful comment, which I have made every attempt to take into account in a revised version of the paper. I also have carefully addressed your further comment in the file.
Best,
Xiuli Sun
Associate Professor of Economics
School of Statistics, Southwestern University of Finance and Economics

Reviewer 4 Report
There are no additional comments on this paper.
Author Response
Dear Reviewer,
I appreciate you for your precious time in reviewing my paper and providing valuable comments. It was your valuable and insightful comments that led to possible improvements in the current version. I welcome further constructive comments if any.
Thank you very much.
Best,
Xiuli Sun
Associate Professor of Economics
School of Statistics, Southwestern University of Finance and Economics